# Near point-of-care HIV viral load testing: Cascade after high viral load in suburban Yangon, Myanmar

**Ni Ni Tun**[1,2]*, **Frank Smithuis**[1,2], **Nyan Lynn Tun**[2], **Myo Min**[2], **Myo Ma Ma Hlaing**[1], **Josefien van Olmen**[3], **Lutgarde Lynen**[4], **Tinne Gils**[4]

**1** HIV/TB, Medical Action Myanmar, Yangon, Myanmar, **2** HIV/TB, Myanmar Oxford Clinical Research Unit, Yangon, Myanmar, **3** Spearhead Research Public Health & Primary Care, University of Antwerp, Antwerp, Belgium, **4** Clinical Sciences, Institute of Tropical Medicine, Antwerp, Belgium

\* nini@mam.org.mm

## Abstract

### Introduction

HIV viral load (VL) testing in resource-limited settings is often centralised, limiting access. In Myanmar, we assessed outcomes according to VL access and the VL cascade (case management after a first high VL result) before and after near point-of-care (POC) VL was introduced.

### Methods

Routine programme data from people living with HIV (PLHIV) on antiretroviral therapy (ART) were used. We assessed the odds of getting a VL test done by year. Attrition and mortality two years after ART initiation were compared between three groups of PLHIV with different access to VL testing using Kaplan-Meier analysis. We compared VL cascades in those with a first VL result before and after near POC VL testing became available. With logistic regression, predictors of confirmed virological failure after a first high VL in the POC era were explored.

### Results

Among 4291 PLHIV who started ART between July 2009 and June 2018, 794 (18.5%) became eligible for VL testing when it was not available, 2388 (55.7%) when centralised laboratory-based VL testing was available, and 1109 (25.8%) when near POC VL testing was available. Between 2010 and 2019, the odds of getting a VL test among those eligible increased with each year (OR: 5.21 [95% CI: 4.95–5.48]). Attrition and mortality were not different in the three groups. When comparing PLHIV with a first VL result before and after implementation of the near POC VL testing, in the latter, more had a first VL test (92% versus 15%, p<0.001), less had a first high VL result (5% versus 14%, p<0.001), and more had confirmed virological failure (67% versus 47%, p = 0.013). Having a first VL ≥5000 copies/mL after near POC implementation was associated with confirmed virological failure (adjusted OR: 2.61 [95% CI: 1.02–6.65]).

**Data Availability Statement:** The dataset for this study includes human research participants and other sensitive information. The authors studied a cohort of vulnerable populations, i.e. people living

with HIV, among whom key populations (female sex workers, men who have sex with men, people who inject drugs). Potentially indirectly identifying information includes age, treatment clinic, sex, year of ART start. Sensitive information, which could lead to discrimination, includes HIV status, viral load results, and key population status. Stigma against HIV persists in Myanmar, and it is higher for key populations, where the epidemic is concentrated. Key populations are also criminalized, and this has worsened during the current situation of political unrest situation in Myanmar. Data is available upon request from Medical Action Myanmar (info@mam.org.mm) for researchers who meet the criteria for access to confidential data.

**Funding:** The authors received no specific funding for this work.

**Competing interests:** All authors have declared that no competing interests exist.

## Conclusion

Near POC VL testing enabled rapid increase of VL coverage and a well-managed VL cascade in Myanmar.

## Introduction

Scaling up antiretroviral therapy (ART) and access to routine HIV viral load (VL) testing are critical to control the HIV epidemic [1]. The World Health Organization (WHO) recommends VL testing to monitor people living with HIV (PLHIV) at six and twelve months after the start of ART and annually thereafter [2]. In case the VL is above 1000 copies/mL, enhanced adherence counselling (EAC) should be performed and VL testing should be repeated after three months. If the second VL is also above 1000 copies/mL despite EAC, virological failure is confirmed and prompt switch to a second-line regimen is indicated [2]. Though VL is regularly measured in developed countries, access to VL tests is still very limited in many low-resource settings, and this leads to delayed switching to second-line ART [3–7]. The mostly centralised VL testing demands sophisticated and expensive facilities, equipment, and skilled technicians, making it difficult and impractical to scale up. Moreover, challenges with blood collection, sample storage, and transportation often lead to delayed results [8, 9].

To expand VL monitoring, the WHO has recommended point-of-care (POC) VL testing in priority groups since 2021 [2]. In the STREAM study, the time to the return of test results and to clinical decision-making following an elevated VL reduced, and retention in care improved, when POC VL testing was compared to centralised VL testing [10]. Near POC VL testing, an approach using testing in laboratories close to, but not inside, treatment facilities, has shown to be feasible and to enable prompt clinical action in seven countries in Sub-Saharan Africa [11]. Compared to centralised testing, near POC VL testing also improved the turn-around time from sampling to results delivery to PLHIV and to action when high VL was identified [11, 12]. For its implementation, inexpensive, low-complexity, and POC assays are needed. The GeneXpert® HIV-1 VL assay (Cepheid, Sunnyvale, CA; GeneXpert), is an in vitro diagnostic test which uses polymerase chain reaction (PCR) technology to quantify HIV-1 in human plasma from PLHIV [13]. Evidence has shown that the GeneXpert results compare well with the widely-used Abbott® HIV-1 m2000 RealTime PCR (Abbott, Chicago, IL) [13–17]. The GeneXpert is suitable for near POC VL testing and was included in the WHO list of prequalified in vitro diagnostics in 2017 [18, 19].

Myanmar is a low-resource setting with an estimated general HIV prevalence of 0.7%, the second-highest in Southeast Asia [20]. In 2022, Myanmar had an estimated 270 000 PLHIV, 11000 new HIV infections, and 6600 AIDS-related deaths [20]. Seventy-six percent of PLHIV received ART, and 72% of them were virally suppressed in 2020 [21]. The epidemic is highly concentrated among key affected populations (KAP); HIV prevalence is 8.3% among female sex workers (FSW), 8.8% among men who have sex with men (MSM), and 19.0% among people who inject drugs (PWID) [20]. Near POC VL testing was introduced in Myanmar in 2017. Until then, VL testing remained limited to centralised laboratory-based testing with Abbott situated in two major cities, Mandalay and Yangon. Hence, access to VL testing remained limited. In 2017, only half of over 7000 newly initiated PLHIV in five regions across the country received a VL test in the first year on ART [7]. Even in six ART clinics in Yangon, a mere 58% of eligible PLHIV had received a VL test in 2019. Challenges with sample transport resulted in long turn-around times between sampling and reception of results [8].

GeneXpert for near POC VL testing was introduced in Yangon in 2017 by the organisation Medical Action Myanmar (MAM). Evidence on implementation of near POC VL testing remains limited [2]. We assessed the proportion of eligible PLHIV receiving a yearly VL in a ten-year period and compared attrition and mortality according to access to VL testing. We compared outcomes along the VL cascade (i.e. case management after a first high VL) in those with a first VL result before or after near POC VL testing was introduced. In the latter group, we also assessed predictors of confirmed virological failure.

## Methods

### Study setting, design and population

The study took place at three MAM clinics in two of Yangon's large suburban slum areas: Hlaingtharya (clinic 1 and 2) and Shwepyitha (clinic 3). These areas are generally inhabited by people with low incomes, such as day labourers, garment factory workers, taxi and long-haul drivers, and FSW. Many people have recently been internally displaced there from rural areas. MAM is a medical organisation providing care for PLHIV in Myanmar since 2009. MAM operates free-of-charge clinics that provide a package of services, including primary health care such as services for non-communicable diseases, reproductive and antenatal health, tuberculosis, and HIV. Details of the HIV treatment project were described previously [22]. In short, HIV clinical care was performed by medical doctors and nurses. Counselling and adherence support was provided during clinic and home visits by trained lay counsellors and outreach adherence supporters. All services provided followed the National AIDS Programme guidelines, based on the WHO recommendations [2, 23–25]. ART was initiated based on a CD4 cell count below 350 cells/mm$^3$ until early 2017, and regardless of CD4 counts afterwards [23, 25, 26]. First-line treatment consisted of a combination of two nucleoside/nucleotide reverse transcriptase inhibitors (stavudine, zidovudine, tenofovir, lamivudine, abacavir) with a non-nucleoside reverse transcriptase inhibitor (efavirenz or nevirapine) or dolutegravir. Tenofovir became available in November 2013 and dolutegravir in December 2018. Second-line regimens were composed of two nucleoside/nucleotide reverse transcriptase inhibitors and one protease inhibitor (atazanavir/ritonavir, lopinavir/ritonavir). The WHO recommended routine VL monitoring guideline was used [2]. Since 2013, two Abbott HIV-1 VL testing platforms were available in Myanmar, placed at National Health Laboratories (NHL) in Mandalay and Yangon. MAM sites sent VL samples to the NHL in downtown Yangon, at a travel time of maximum two hours by car from the clinics. The Yangon NHL received samples from all public facilities in the country's South. Due to limited availability of qualified staff, VL testing was performed only three days a week. In practice, a monthly maximum of two samples per MAM clinic could be sent for VL testing, results turn-around-time could extend over a month, and many samples got lost. From January 2017, the three MAM clinics shared one near POC GeneXpert, installed in a small laboratory within 30 minutes by motorbike from each clinic and operated by trained lay personnel. Blood samples were collected daily from clinics and sent to the laboratory. Results were provided by phone to the clinic team about two hours after the sample arrived at the testing site, and on paper the next morning. We retrospectively analysed routinely collected data of PLHIV enrolled on ART in the three MAM clinics between July 2009 and June 2018. The database was closed in December 2019.

We assessed compliance with yearly VL testing in PLHIV eligible to have a VL test done (alive on ART for minimum six months) in a certain year between 2010 and 2019. We compared attrition and mortality during the first two years on ART between three groups, based on the availability of VL testing when the PLHIV became eligible for VL testing. Group 1 consisted of PLHIV initiated on ART between July 2009 and June 2012, who became eligible before VL

testing was available ("No VL"). Group 2 included PLHIV initiated on ART between July 2012 and June 2016, who became eligible for VL testing after introduction of laboratory-based centralised VL and before near POC VL testing ("Laboratory-based VL"). Group 3 included PLHIV initiated on ART between July 2016 and June 2018, who became eligible for VL testing after the introduction of near POC VL testing ("Near POC VL"). We compared the VL cascade between those with a first VL result before and after near POC VL testing was implemented. We assessed risk factors associated with confirmed virological failure after EAC in the latter group.

## Study variables

The study used routine programme data collected from standardised patient forms in hard copy, designed for the use of the FUCHIA (Follow-Up Care of Clinical HIV infection and AIDS) software. Values recorded during the ART initiation visit were considered baseline for age, sex, marital status, employment, KAP status (defined as self-identifying as FSW, MSM, and/or PWID), clinic, place of origin, mode of entry, WHO stage, and tuberculosis co-infection. We defined CD4 cell count at ART initiation as the measurement taken closest to the date before ART initiation. Compliance with yearly VL testing was defined as the proportion of PLHIV on ART who had at least one VL done among those who were alive on ART for at least six months during the respective year. A first VL >1000 copies/mL at least six months after ART initiation was defined as first high VL. A first VL of ≤1000 copies/mL was defined as suppressed VL. Confirmed virological failure was defined as a follow-up VL >1000 copies/mL after the first high VL and EAC. A follow-up VL ≤1000 copies/mL was defined as re-suppressed VL. EAC is a special form of counselling for ART patients with a first high VL in which the counsellor and patient go through the identification of barriers to adherence and strategies to overcome them. To construct the VL cascade, we assessed the proportion of ART patients who were eligible and who underwent VL monitoring, had a first high VL, had a follow up VL test, had confirmed virological failure, were switched to second-line ART, had a VL test after the switch, and had a VL <1000 copies/mL after the switch [24]. Dead was confirmed death for any reason during the observation period. Lost to follow up (LTFU) was defined as a delay of more than three months between an expected and actual visit. The LTFU date was the last date of the clinic visit. A patient transferred to a non-MAM ART centre during the observation period was defined as transferred out (TO). Those receiving ART at a MAM clinic until the end of the observation period were considered alive on ART. Attrition included dead and LTFU.

## Data collection and management

Any identified PLHIV at a MAM clinic received a numeric code for the FUCHIA database. Medical doctors kept paper-based patient files updated, which were routinely entered into the FUCHIA software by trained data clerks. No directly identifying information was recorded in the FUCHIA database and the FUCHIA code was used for any medical communication, including linking laboratory results and EAC data to patient files. Included PLHIV received a separate study code before data were extracted into Microsoft Excel to create the study database. The key to link study codes with FUCHIA codes was password protected, only accessible to the first author. In case of discrepancies between the FUCHIA database and patient files, original records were verified, and corrections made. Data cleaning and verification was conducted by the first author.

## Data analysis

Categorical variables were summarised with frequencies and proportions, whilst continuous variables were summarised with medians and interquartile ranges (IQR) or means and

standard deviations, as appropriate. We assessed the association between increasing year of eligibility and getting a VL when eligible testing with a logistic regression. Kaplan-Meier techniques and a log-rank test were used to compare time to attrition and death in the three groups according to VL access. Follow-up time was started the day PLHIV initiated ART and ended on the date of the outcome or at database closure (31 December 2019) for those alive on ART. ART patients transferred out were censored the day they were transferred. We compared the VL cascade of PLHIV who received a first VL result before and after introduction of near POC VL testing in 2017 using Chi-squared or Fisher's exact test, as appropriate. We used stepwise multivariable logistic regression from a saturated model to explore predictors of having a confirmed virological failure after EAC among patients who received first VL results after the introduction of near POC VL testing. Age and sex were retained in the multivariable model, as well as variables with p-values ≤0.05. Odds ratios (OR) and 95% confidence intervals (CI) were presented. All analyses were done using Stata 16.1 (Stata Corp, College Station, TX).

### Ethics statement

As this was a retrospective analysis, it would have been impractical to collect informed consent. Ethical clearance was obtained from the Institutional Review Board of the Institute of Tropical Medicine, Antwerp, Belgium (Reference: IRB/AB/AC/058 1363/20). A consent waiver and an exemption from ethical reviews were granted by the Oxford Tropical Research Ethics Committee of the University of Oxford. Since the military coup in February 2021 in Myanmar, the Ministry of Health and the Institutional Review Board of the Department of Medical Research are barely functional.

## Results

### Baseline characteristics

Baseline characteristics of 4291 PLHIV who initiated ART between July 2009 and June 2018, stratified by access to VL testing, are shown in Table 1.

Overall, 541 reported to be FSW (12.6%), 170 MSM (4.0%), and 85 PWID (2.0%). Among those without access to VL testing, access to laboratory-based VL, or near POC VL testing when becoming eligible, respectively, 53 (6.7%), 289 (12.1%), and 199 (17.9%) reported to be FSW, 10 (1.3%), 77 (3.2%), and 83 (7.5%) to be MSM, and 8 (1.0%), 57 (2.4%), and 20 (1.8%) to be PWID.

### Compliance with yearly viral load testing

Before 2017, overall, 432 VL tests were carried out among 2902 eligible patients, or 0.2 VL tests per person. After 2017, 7140 VL tests were done among 3561 eligible patients, or 2.0 VL per person. Compliance with yearly VL tests in eligible patients between 2010 and 2019 was 0% (0/30), 0% (0/289), 0% (0/443), 4.6% (49/1055), 5.6% (89/1579), 2.4% (49/2068), 8.2% (204/2487), 32.6% (931/2858), 91.9% (2786/3033), and 94.5% (2649/2804), respectively for each year (Fig 1). For each extra year, the odds of getting a VL test significantly increased: OR: 5.21 [95% CI: 4.95–5.48]. The strength of the association did not change when excluding years in which no VL testing was performed.

### Comparison of attrition and mortality

When comparing two-year attrition and mortality in Kaplan-Meier analysis, there was no significant difference in attrition or mortality between the three groups or between the two groups who became eligible in the time period when VL testing was available (Fig 2).

**Table 1. Baseline characteristics by access to VL testing of PLHIV initiated on ART between June 2009 and July 2018 in three clinics in Yangon, Myanmar.**

| | | | Total | | Access to VL | | | | | |
|---|---|---|---|---|---|---|---|---|---|---|
| | | | | | No VL | | Laboratory-based VL | | Near point-of-care VL | |
| Variable | | | N = 4291 | | n = 794 | | n = 2388 | | n = 1109 | |
| | | | n | % | n | % | N | % | n | % |
| Median age (IQR, years) | | | 34 | [28–40] | 34 | [29–40] | 34 | [28–40] | 33 | [28–39] |
| Age category (years) | | <24 | 351 | 8% | 44 | 6% | 190 | 8% | 117 | 11% |
| | | 25–49 | 3635 | 85% | 713 | 90% | 2005 | 84% | 917 | 83% |
| | | >50 | 305 | 7% | 37 | 5% | 193 | 8% | 75 | 7% |
| Sex | | female | 2017 | 47% | 336 | 42% | 1121 | 47% | 560 | 50% |
| | | male | 2274 | 53% | 458 | 58% | 1267 | 53% | 549 | 50% |
| TB at initiation | | yes | 1409 | 33% | 187 | 24% | 921 | 39% | 301 | 27% |
| | | no | 2882 | 67% | 607 | 76% | 1467 | 61% | 808 | 73% |
| Median CD4 (IQR, cells/mm$^3$) | | | 185 | [70–311] | 162 | [58–252] | 179 | [70–303] | 240 | [82–403] |
| CD4 category (cells/mm$^3$) | | <200 | 2266 | 53% | 479 | 60% | 1302 | 55% | 485 | 44% |
| | | ≥200 | 1989 | 46% | 301 | 38% | 1075 | 45% | 613 | 55% |
| | | unknown | 36 | 1% | 14 | 2% | 11 | 0% | 11 | 1% |
| WHO stage | | I/II | 1417 | 33% | 137 | 17% | 770 | 32% | 510 | 46% |
| | | III/IV | 2874 | 67% | 657 | 83% | 1618 | 68% | 599 | 54% |
| ART history | | yes | 185 | 4% | 49 | 6% | 100 | 4% | 36 | 3% |
| | | no | 4106 | 96% | 745 | 94% | 2288 | 96% | 1073 | 97% |
| Part of key population† | | yes | 796 | 19% | 71 | 9% | 423 | 18% | 302 | 27% |
| | | no | 2888 | 67% | 500 | 63% | 1616 | 68% | 772 | 70% |
| | | unknown | 607 | 14% | 223 | 28% | 349 | 15% | 35 | 3% |
| Origin | | Yangon | 523 | 12% | 112 | 14% | 341 | 14% | 70 | 6% |
| | | outside Yangon | 3765 | 88% | 682 | 86% | 2047 | 86% | 1036 | 93% |
| | | unknown | 3 | 0% | 0 | 0% | 0 | 0% | 3 | 0% |
| Marital status | | married | 1507 | 35% | 299 | 38% | 869 | 36% | 339 | 31% |
| | | not married | 2783 | 65% | 495 | 62% | 1519 | 64% | 769 | 69% |
| | | unknown | 0 | 0% | 0 | 0% | 1 | 0% | 0 | 0% |
| Employment | | employed | 3159 | 74% | 557 | 70% | 1766 | 74% | 836 | 75% |
| | | unemployed | 1132 | 26% | 237 | 30% | 622 | 26% | 273 | 25% |
| Clinic | | clinic 1 | 2928 | 68% | 769 | 97% | 1605 | 67% | 554 | 50% |
| | | clinic 2 | 626 | 15% | 1 | 0% | 308 | 13% | 317 | 29% |
| | | clinic 3 | 737 | 17% | 24 | 3% | 475 | 20% | 238 | 21% |

† Part of key population includes self-reported female sex workers, men-who-have-sex-with-men, people-who-inject-drugs

VL = viral load, ART = antiretroviral therapy, IQR = interquartile range, PLHIV = people living with HIV, TB = tuberculosis, WHO = World Health Organization

## VL cascade after a first high viral load

Among PLHIV eligible for VL testing, 0% (0/732) had received a first viral load before 2013, 14.6% (370/2532) received a test between 2013 and 2017, and 91.9% (2945/3205) received a test after 2017, respectively. The cascades after a first high VL for PLHIV who received a first VL test during the 2013–2017 period and after 2017 are shown in Table 2.

For those with a first VL between 2013 and 2017, and those after 2017, respectively, the median time to a first VL was 39 (IQR: 24–49) and 35 months (IQR: 17–53). Among PLHIV with a first VL>1000 copies/mL between 2013 and 2017, and those after 2017, respectively, 49% (26/53) and 28% (41/125) had a suppressed follow-up VL (p = 0.004).

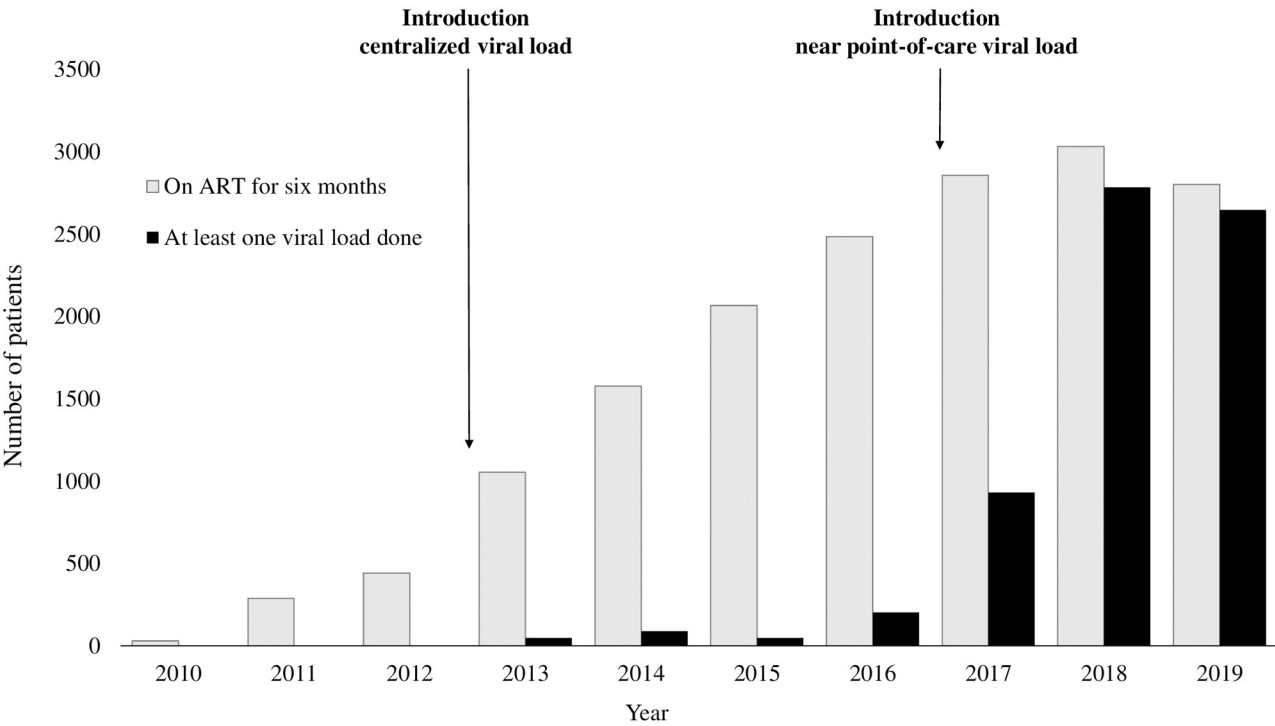

**Fig 1. Compliance with yearly VL testing among eligible PLHIV in three clinics in Yangon, Myanmar from 2010 to 2019.** ART = antiretroviral therapy, PLHIV = people living with HIV/AIDS.

### Predictors of confirmed virological failure

Predictors for confirmed virological failure after EAC among PLHIV with a first VL result after introduction of near POC VL testing are shown in Table 3. Among 125 PLHIV who underwent EAC and had a follow-up VL, having a first high VL result of ≥5000 copies/mL was the only factor significantly associated with confirmed virological failure in both univariate and multivariate analysis (adjusted OR: 2.61 [95% CI: 1.02–6.65]).

## Discussion

This study is the first assessment of the VL cascade after POC VL testing was introduced in Myanmar. Compliance with yearly VL monitoring improved significantly over ten years, when first laboratory-based VL and later near POC VL testing was introduced. However, two-year attrition and mortality was similar in groups with different access to VL testing. We report good results along the VL cascade after the introduction of near POC VL testing. In those with a first VL at near POC, compared to before, more PLHIV had a VL test done (p<0.001), of whom less had a first high VL (p<0.001), but among those with a second VL, more had confirmed virological failure (p = 0.013). Having a first VL of 5000 copies/mL or higher was associated with confirmed virological failure in group with a first VL result after the introduction of near POC VL testing.

Less than 10% of those eligible received a yearly VL until 2016, due to limited testing capacity and high sample load at the central laboratories. Factors contributing to the fast increase after 2017 were reduction in results turn-around-time from about a month to a day, increased staff experience, and refresher trainings at near POC VL testing introduction. It is likely that

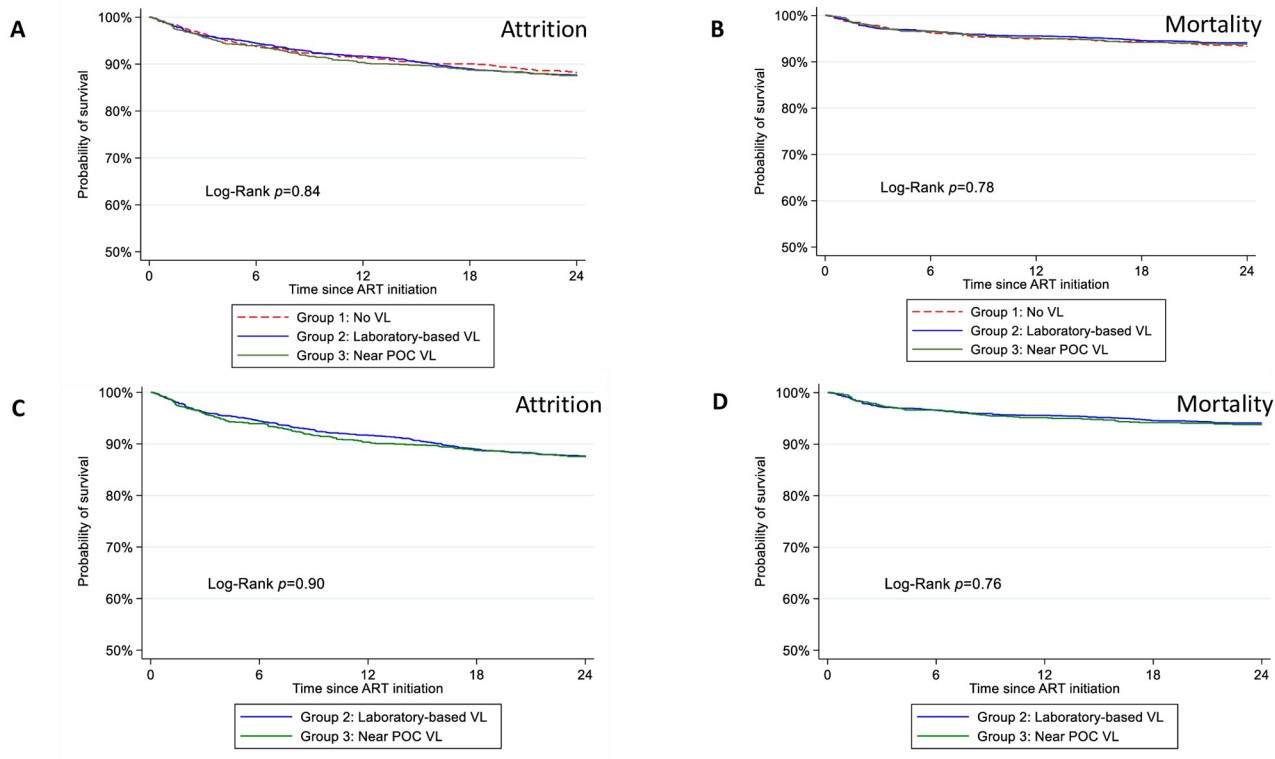

**Fig 2. Comparing outcomes on ART among PLHIV with different access to VL testing.** Panel A, C present two-year attrition (dead, lost-to-follow-up) and panel B, D two-year mortality between PLHIV on ART who when becoming eligible had no access to VL testing (Group 1: No VL), access to laboratory-based VL testing (Group 2: Laboratory-based VL) or access to near POC VL testing (Group 3: Near POC VL). Panels A and B include all three groups while Panels C and D only. compare Groups 2 and 3. ART = antiretroviral therapy, PLHIV = people living with HIV/AIDS, POC = point of care, VL = viral load.

**Table 2. Comparison of cascade after a first high VL before and after near POC VL testing was implemented.**

| Cascade step | Laboratory-based VL between 2013 and 2017 | | Near-POC VL after 2017 | | p-value* |
|---|---|---|---|---|---|
| | **n** | **%** | **n** | **%** | |
| Eligible for first viral load | 2532 | | 3205 | | |
| First viral load done | 370 | 14.6% | 2945 | 91.9% | <0.001 |
| First viral load >1000 copies/mL | 53 | 14.3% | 149 | 5.1% | <0.001 |
| Follow-up viral load after EAC done | 49 | 92.5% | 125 | 83.9% | 0.165† |
| Follow-up viral load >1000 copies/mL | 23 | 46.9% | 84 | 67.2% | 0.013 |
| Switch to second-line ART | 21 | 91.3% | 69 | 82.1% | 0.355† |
| Viral load after switch done‡ | 21 | 100.0% | 54 | 94.7% | 0.559† |
| Viral load after switch ≤1000 copies/mL | 20 | 95.2% | 47 | 87.0% | 0.429† |

* Chi-square or Fisher's Exact test (†)

‡ after 2017 the proportion is 54/57, as 57 had enough follow-up time to observe viral load after switch

VL = viral load, ART = antiretroviral therapy, EAC = enhanced adherence counselling, POC = point-of-care

**Table 3. Factors associated with confirmed virological failure in ART patients with enhanced adherence counselling after a first high near POC VL in Yangon, Myanmar.**

| Variable | n | % | Univariate analysis | | | Multivariate analysis | | |
|---|---|---|---|---|---|---|---|---|
| | | | OR | 95% CI | p | aOR | 95% CI | p |
| Age | | | 1.00 | [0.95–1.04] | 0.86 | 1.00 | [0.96–1.05] | 0.91 |
| Sex | | | | | | | | |
| Male | 72 | 58% | ref | | | | | |
| Female | 53 | 42% | 0.59 | [0.28–1.25] | 0.17 | 0.67 | [0.31–1.47] | 0.32 |
| Tuberculosis | | | | | | | | |
| No | 65 | 52% | ref | | | | | |
| Yes | 60 | 48% | 2.00 | [0.93–4.31] | 0.08 | | | |
| Tuberculosis at first VL | | | | | | | | |
| No | 118 | 94% | ref | | | | | |
| Yes | 7 | 6% | 3.08 | [0.36–26.4] | 0.31 | | | |
| CD4 count (cells/mm$^3$) available | 125 | 100% | 1.00 | [1.00–1.00] | 0.50 | | | |
| CD4 category (cells/mm$^3$) | | | | | | | | |
| <200 | 92 | 74% | ref | | | | | |
| ≥200 | 33 | 26% | 0.81 | [0.35–1.86] | 0.61 | | | |
| WHO stage | | | | | | | | |
| I/II | 27 | 22% | ref | | | | | |
| III/V | 98 | 78% | 1.27 | [0.52–3.09] | 0.60 | | | |
| ART history before baseline | | | | | | | | |
| Yes | 4 | 3% | ref | | | | | |
| No | 121 | 97% | 6.55 | [0.66–65.0] | 0.11 | | | |
| Part of key population † | | | | | | | | |
| No | 81 | 65% | ref | | | | | |
| Yes | 42 | 34% | 0.85 | [0.39–1.87] | 0.69 | | | |
| Place of origin | | | | | | | | |
| Yangon | 116 | 93% | | | | | | |
| Outside Yangon | 9 | 7% | 1.77 | [0.35–8.94] | 0.49 | | | |
| Marital status | | | | | | | | |
| Married | 78 | 62% | ref | | | | | |
| Not married | 47 | 38% | 1.28 | [0.59–2.74] | 0.53 | | | |
| Employment | | | | | | | | |
| Unemployed | 30 | 24% | ref | | | | | |
| Employed | 95 | 76% | 1.83 | [0.79–4.26] | 0.16 | | | |
| Clinic site | | | | | | | | |
| Clinic 1 | 75 | 60% | ref | | | | | |
| Clinic 2 | 31 | 25% | 1.54 | [0.62–3.81] | 0.35 | | | |
| Clinic 3 | 19 | 15% | 3.36 | [0.90–12.56] | 0.07 | | | |
| Years between ART initiation and first VL | | | | | | | | |
| <3 years | 89 | 71% | ref | | | | | |
| ≥3 years | 36 | 29% | 0.58 | [0.26–1.29] | 0.18 | | | |
| First VL | | | | | | | | |
| <5000 copies/mL | 25 | 20% | ref | | | | | |
| ≥5000 copies/mL | 100 | 80% | 2.79 | [1.14–6.84] | 0.03 | 2.61 | [1.02–6.65] | 0.05 |

† excluding two with missing data. Part of key population includes self-reported female sex workers, men-who-have-sex-with-men, people-who-inject-drugs

Variables reported those measured at ART initiation, unless specified

aOR = adjusted odds ratio, ART = antiretroviral therapy, OR = odds ratio, VL = viral load, POC = point-of-care, WHO = World Health Organization

the improved flow of VL testing and results increased staff motivation to perform VL testing. The relatively simple GeneXpert procedure allowed task shifting to lay workers. Uninterrupted supply of cartridges was also secured, which had been a challenge with centralised VL testing. A reduction of results turn-around-time was also observed after the introduction of near POC VL testing in seven countries in Sub-Saharan Africa [11].

While the proportion of patients presenting with advanced HIV decreased over time, it was still 44% in those presenting after 2017. To reduce mortality in this group, a package of care including a POC test to detect CD4 counts below 200 cells/mm$^3$, TB- lipoarabinomannan, and cryptococcal antigen tests is recommended [27]. Only 19% of PLHIV self-identified as KAP, which is likely underreported because of persistent stigma. The proportion KAP increased with time. This might be due to clinic staff gained experience in exploring KAP status among PLHIV and or stigma might have reduced over time.

We found no difference in attrition or mortality in the first two years after ART initiation between those eligible for VL testing when VL testing was not yet available, those when centralised VL testing was introduced, and those when near POC VL testing was available, nor between the latter two groups. In the STREAM randomised controlled trial, retention in care and viral suppression was significantly better in a group receiving POC VL testing with task-shifting compared to laboratory-based VL testing. However, retention at any clinic 12 months after ART initiation was also not found to be different between the groups [10]. In Uganda, access to VL testing also did not improve survival when participants were randomised to different VL monitoring strategies [28]. We did not correct for confounders in this analysis, and two years may also be too short to witness an effect of improved access to VL testing in a real-life setting. Retention in care was high before VL introduction in our cohort and remained high when laboratory-based VL testing was available. Measures to improve medicine adherence and retention in care should thus be continued and strengthened, regardless of the VL platform used.

The VL cascade was well managed after near POC VL testing became available. Ninety-two percent among eligible PLHIV had a VL test done. This was significantly higher compared to the pre-POC era in the same programme (15%). A review conducted in low- and middle-income countries reported VL testing coverage of 74% (median, IQR: 46–82%) [29]. The time to a first VL was long, also for those with a first VL result after near POC VL testing became available (35 months, IQR: 17–53). This is expected, as it includes a backlog of PLHIV with no or limited access to VL testing before. Despite the long waiting time for some for a first VL test, 95% had a VL <1000 copies/mL in the near POC group compared to 86% before, when VL was more targeted and access more limited. In comparison, proportions of PLHIV with supressed first VL in Africa and Asia, including Myanmar, ranged between 72–91% [3, 8, 30–34]. As treatment adherence was probably good in our cohort, the long delay in VL testing did not seem to have a negative impact on VL results [22, 35].

After a first high VL result, 93% and 84% among those with a first VL test before and after near POC had a follow-up VL done, and 53% and 33% re-suppressed, respectively. Patients with re-suppression benefitted from strengthening adherence and avoiding unnecessary switch to second-line regimens. Only 66% (median, IQR: 38–77%) of those eligible had a follow-up VL in low- and middle-income settings in a recent review [29]. Another review reported a viral re-suppression of 53% [36]. Among PLHIV with a first high VL after near POC implementation, less had a suppressed follow-up VL documented compared to those who had a first high VL before. This could be explained by the fact that due to the limited laboratory-based testing capacity, targeted VL testing prioritized PLHIV with demonstrated adherence problems. This population could thus have benefitted more from the EAC sessions. In the near-POC era, all back-log cases who had never had a VL were tested, among whom more

might have experienced true treatment failure. While near POC technology can support adherence through rapid turnover of VL results, other adherence strengthening measures remain essential to achieve viral suppression and retain patients in care. In our study, 91% and 82% of those with confirmed virological failure were switched to second-line ART, in those with access to VL testing before and after near POC VL testing, respectively. Much lower proportions switched were reported in systematic review 45% (mean, IQR: 35–71) and also by Médecins Sans Frontières in 10 African sites (10–68%) [29, 37]. In Myanmar, however, high proportions switched to second-line ART (85%) were also reported in another cohort supported by a medical organisation, which probably benefitted from more resources compared to public services [7].

In the near POC VL testing period, a first high VL result of ≥5000 copies/mL was the only factor significantly associated with confirmed virological failure (aOR: 2.61, 95% CI:1.02–6.65). In Zimbabwe and Ethiopia PLHIV with a first VL of respectively ≥5000 copies/mL and ≥10000 copies/mL also had higher odds of virological failure after EAC [34, 38]. Because of lack of drug resistance testing, it is difficult to ascertain whether PLHIV experience virological failure because of continuing adherence problems or because of resistance. In general, a resistant virus is associated with reductions in fitness, and thus we expect lower plasma VL in ART patients who are failing due to resistance. However, the relationship between adherence, drug resistance and plasma VL level is complex and differs depending on ART regimen used and also the frequency with which VL is measured [39, 40]. In the era of dolutegravir-based regimens, with a high resistance barrier, it would be interesting to study whether these factors continue to be important. Among PLHIV with high VL, the proportion resulting from non-adherence is likely to increase in the dolutegravir era. Hence, near POC VL testing remains relevant to trigger faster EAC and subsequent management.

This study has several limitations. In this retrospective analysis, there could be unmeasured confounding factors. The sample size for regression was small, allowing for limited conclusions based on the results. Our clinics are managed by a private medical organisation receiving external funding, and results might not be generalizable to other public facilities. However, POC sampling and analysis were managed by lay workers, with skills easily transferable to public health facilities. Even though experiences showed that most results were received by phone about two hours after samples arrived at the testing site and paper results on the next morning, the actual turn-around-time of near POC was not systematically recorded. Our study also has important strengths. We are the first to document the impact of near POC VL testing on VL management in Myanmar, and in Southeast Asia. We present real-life results from ten years of programme implementation in a large cohort in a resource-limited setting. The study database was thoroughly checked for discrepancies with the source documents by the first author. Despite the retrospective design, we had very little missing data. We are also confident the database reflects the data collected from participants.

We respond to a research gap identified by the WHO addressing implementation of POC VL testing. The WHO conditionally recommends POC VL testing, adapted to the capacity of health facilities, and while giving priority to populations at risk [2]. The usefulness of POC testing indeed depends on the local HIV burden and the sample load. We provided near POC VL testing, shared among three clinics in an area with a general HIV prevalence of 0.7% [20]. Compared to POC testing, near POC testing is cheaper and easier to scale-up. While turn-around-time could be shorter in POC testing compared with near POC testing, most results in our setting were received telephonically two hours after sample arrival, allowing prompt clinical action. In addition, we optimised human and financial resources by sharing the instrument between three clinics rather than one. Thus, near POC VL testing can balance costs versus needs in low- or medium-burden HIV settings, with limited resources and access challenges.

After 2019, access to services for PLHIV deteriorated rapidly in Myanmar due to COVID-19 and later political turmoil. The first author was strongly involved in mitigating the impact of these crises, delaying this analysis. The positive impact of improved VL monitoring and more robust ART regimens on viral suppression could be countered by unplanned treatment interruptions due to poor health service access in this period. Effectively, health care workers strike, arrests of and threats to health care staff, and armed conflict resulted in a gap in public health care. In addition, security issues restrict patients' freedom to travel to health facilities [41]. Near POC VL provides an opportunity to mitigate damage to VL coverage during these crises, as services that are faster and closer to patients could limit service interruption. Future research should show if near POC VL testing has this potential. In other contexts, with similar constraints in access and resources, near POC could provide an acceptable alternative to POC testing, improving time to a VL test and subsequent management while limiting the burden on primary care facilities.

## Conclusion

After near POC VL testing was introduced, VL coverage improved significantly in our setting, preceding a well-managed VL cascade. In resource-constraint settings where access to health care is challenging, near POC VL testing provides a pragmatic alternative to POC VL testing to help control the HIV epidemic.

## Supporting information

**S1 File.**
(DOCX)

## Acknowledgments

We would like to thank study participants, the medical and data teams from Medical Action Myanmar, and the Myanmar Oxford Clinical Research Unit, Yangon, Myanmar.

## Author Contributions

**Conceptualization:** Ni Ni Tun, Lutgarde Lynen, Tinne Gils.

**Data curation:** Ni Ni Tun, Nyan Lynn Tun, Myo Min, Myo Ma Ma Hlaing, Tinne Gils.

**Formal analysis:** Ni Ni Tun, Nyan Lynn Tun, Tinne Gils.

**Funding acquisition:** Frank Smithuis.

**Methodology:** Ni Ni Tun, Lutgarde Lynen, Tinne Gils.

**Project administration:** Ni Ni Tun.

**Resources:** Ni Ni Tun, Frank Smithuis.

**Supervision:** Ni Ni Tun, Frank Smithuis, Josefien van Olmen, Lutgarde Lynen, Tinne Gils.

**Validation:** Ni Ni Tun, Frank Smithuis, Josefien van Olmen, Lutgarde Lynen, Tinne Gils.

**Visualization:** Ni Ni Tun, Lutgarde Lynen.

**Writing – original draft:** Ni Ni Tun.

**Writing – review & editing:** Ni Ni Tun, Frank Smithuis, Josefien van Olmen, Lutgarde Lynen, Tinne Gils.

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
