## [Decision Letter · Decision Letter 0]

4 Nov 2022

PONE-D-22-20673Near point-of-care HIV viral load testing: Uptake and utilization in suburban Yangon, MyanmarPLOS ONE

Dear Dr. Tun,

Thank you for submitting your manuscript to PLOS ONE. After careful consideration, we feel that it has merit but does not fully meet PLOS ONE’s publication criteria as it currently stands. Therefore, we invite you to submit a revised version of the manuscript that addresses the points raised during the review process.

Reviewer 2 has requested significant changes that are necessary to clarify what was done in this study and to raise the level of usefulness of the data presented. In addition, be sure to have the grammar checked before submitting a revision. 

We look forward to receiving your revised manuscript.

Kind regards,

Julie AE Nelson, PhD

Academic Editor

PLOS ONE

Journal Requirements:

“NO”

“NO”

Reviewers' comments:

Reviewer's Responses to Questions

**Comments to the Author**

1. Is the manuscript technically sound, and do the data support the conclusions?

Reviewer #1: Yes

Reviewer #2: Partly

2. Has the statistical analysis been performed appropriately and rigorously? 

Reviewer #1: Yes

Reviewer #2: I Don't Know

3. Have the authors made all data underlying the findings in their manuscript fully available?

Reviewer #1: Yes

Reviewer #2: No

4. Is the manuscript presented in an intelligible fashion and written in standard English?

Reviewer #1: Yes

Reviewer #2: No

5. Review Comments to the Author

Reviewer #1: This was a very nicely written manuscript that touches on an important topic in a clear, structured way.

Major Comments

1. The Abstract was a little unclear regarding the time of first viral load after initiation. Line 42-43 indicates 2.8 years after introduction of near POC VL, but then line 44 says 0.9 years in those becoming eligible after near POC VL. 3.7 years before near POC VL introduction. It would be helpful to clarify this as I’m guessing the intention was for the three different groups and/or one was an overall. Additionally, the results only includes two of those three timings and would benefit from including all (lines 263-266).

2. Please also review the patient numbers. The results were a bit unclear here. The abstract and results list 5271 PLHIV started ART between July 2009-December 2019. Line 230, however, lists 4291 PLHIV initiated ART. Then, Group 1 (794), Group 2 (2386), Group 3 (1111). Line 244 repeats 4291 initiated ART during the same timeframe and then indicates 3205 were eligible after near POC introduction (Group 3, earlier with n=1111). Finally, line 263, presents 945 eligible after near POC. It would be very helpful to ensure accuracy throughout and perhaps clearer explanations. The use of ‘Group 1’, ‘Group 2’, etc might also help.

3. It would be helpful to discuss whether clinicians and facilities received more clinical training and/or viral load training refreshers at near POC VL implementation and that this could partly be a cause of the improvements seen.

4. Some references could use an update. References 2 and 3 are on older WHO guidelines (2013 and 2016). These are referenced in the introduction and methods. These could be updated to 2021 as the referenced recommendations have all been carried forward and are the relevant, present guidelines.

5. The WHO treatment monitoring algorithm should be updated: testing is suggested at 6 months, 12 months, and yearly thereafter. The timing at 12 months post-ART initiation seems to be missing.

6. In the discussion, it would be helpful to touch on the possible role of near POC VL with uptake and implementation of DTG-based regimens, given the higher barrier to drug resistance and higher viral suppression rates. Further, please discuss how the current political situation in Myanmar could impact this strong system built.

7. It would be helpful to discuss the previous NHL testing strategy. Why would they only accept 1-2 samples per clinic per month? Are they only testing one day a month? Is it due to staff, funding? This seems incredibly low and a waste of available resources.

8. In the discussion, please touch on data quality and the confidence you have in its accurateness, not just completeness.

9. Please touch on how data were collected to ensure confidentiality – given that some patient identifying information was necessary to link across databases and files.

10. In the discussion, it would be quite imperative to discuss the health of those initiating ART. Table 1 indicates that nearly 50% have advanced HIV disease and over 60% have WHO stage 3 or 4. Further, it was suggested that the majority of PLHIV in Myanmar identified as key populations, yet Table 1 has only 21%. Please explain and discuss this discrepancy.

11. For the results on lines 269-272 and in the discussion, it would be good to indicate that the value of the first viral load was the ‘only’ significantly associated metric of those assessed. Further, it would be good to include the OR in the text as well. Nearly 3x is important and significant.

12. In lines 309 and elsewhere in the discussion, it may be important to note that it isn’t expected that a technology or device can do it all and/or impact adherence, retention, etc. Those must still be strengthened, etc.

13. Line 326-327 is a little troubling, suggesting that adherence counseling not be necessarily for those with a first high VL. However, 36% of PLHIV in this study re-suppressed after adherence and likely were much better off staying on 1st line then switching unnecessarily. Further, this (EAC and switch hesitation) is likely to be particularly important with DTG scale-up.

14. When discussing the recent WHO recommendation on POC VL (lines 356+), it may be useful to note that they are conditional based on high volumes and limited device throughput, particularly in high burden settings. To this point, it would be helpful to highlight the differences and challenges with POC vs near POC (ie same day versus not same day, need for clear follow-up structures and resources to maximize near POC use, possible improvements in TAT and clinical decision-making vs lab-based). Particularly as the model developed at these sites in Myanmar is clearly incredibly strong and well-resourced given impressive retention, etc.

Minor Comments

1. Line 49, should this be ‘ninety-three’?

2. It would be helpful to reference lines 74-76.

3. For references on lines 82-83, it would be good to also reference Sacks et al: https://pubmed.ncbi.nlm.nih.gov/31274537/.

4. Line 101 is missing ‘in’ prior to ‘2017’.

5. Line 127: Abbott should be written with two ‘t’s.

6. Line 163, it is unclear what ‘KAP status’ means.

7. Please look at line 193-194, it seems incomplete.

8. The sentence from line 208-210 seems irrelevant and could be removed.

9. Line 311, ensure inclusion of ‘near POC’.

10. Lines 329-330 should also reference MSF’s work on the VL cascade: https://msfaccess.org/sites/default/files/MSF_assets/HIV_AIDS/Docs/AIDS_report_Part_1_MakingViralLoadRoutine_MSF_VL_Programmatic%20Report_Web_2016_ENG.pdf (Table 1).

11. Lines 340-341 seemed a bit out of place with that paragraph.

12. Line 347, I might suggest ‘…generalizable to other public facilities.’

13. Line 360 likely needs an ‘and’ after ‘COVID-19’.

Reviewer #2: In this article by Tun et al., the authors describe implementation of near point-of-care quantitative HIV RNA testing in three clinics in Burma. The implementation of POC NAT for viral load monitoring is critical for HIV care and this manuscript presents important results. However, I found the analysis and presentation to be confusing and difficult to follow.

It is unclear why baseline characteristics are being presented only after other data and only for a subset of the population. The first results presented should be the baseline characteristics of the populations for all three time periods, divided first into people initiating ART during the three time periods.

Then I would suggest the main analyses to be presented are:

1) How many total HIV RNA tests were done before and after implementation of POC NAT, and what was the number per for patient on ART (note that this will be different than the baseline groups, because some people will carry forward into the next time period).

What was the actual turn-around-time of POC NAT?

I cannot find either of these results.

2) What % of eligible persons received their first VL testing in each of the three periods (<2013, 2013-2017, >2017)?

These are the data shown in Fig 2 and I believe in lines 219-224. If these are the same data, they should be labeled on the actual figure and can be removed from text.

Is there an explanation for why the number was so much lower in 2017 c/w 2019? That explanation should be included in the Discussion.

The comparison (line 223) should not be 2010-19 v 2013-19 v 2017-19, and there is probably a more informative statistical test that could be used here to show the magnitude of change.

3) Comparing the two time periods (2013-2017 and after 2017):

Of persons who had a first VL>1000, what % had a second test?

If they had a second test, what % had a second VL >1000

Of persons with a second VL>1000, what % had an ART change

Of persons with an ART change, what % had a follow-up VL < (under) 1000.

I do not see any data on the 2013-17 period.

4) Among people starting ART in 2017 who had access to POC NAT, was there lower mortality compared to historical controls (perhaps people starting ART in 2013, when VL testing became available).

This looks like it would be the data shown in Figure 3, although I am not clear on what definition of “attrition” is being used – is it discontinuation of ART? Transfer to another clinic? Lost to follow-up? Mortality? A combination endpoint of all of the above? This should be clarified in the Methods section. Mortality is, of course, the endpoint that matters most.

It would be helpful to label the figure legend so that the reader can easily understand what Groups 1-3 represent.

Note that these analysis groups will be different from those in the above (persons eligible for first VL) because there will be survivor bias among persons who were started on ART in the prior period and who survived without NAT into the next period (presuming there is a universal recommendation to start ART regardless of CD4 count and that persons initiating ART in the two periods do not significantly differ in baseline characteristics)

Minor comments:

) I am not sure that this manuscript describes “uptake”, since it is not clear which patients were offered POC NAT and who declined.

) Please note somewhere (probably in the methods – around lines 121-126) whether the ART start guidelines changed (e.g. based on CD4 versus universal testing) and whether HIV treatment options changed over the course of the study period, as improvements in ART would confound the analysis.

) Methods, line 137 – please provide explanation for why the database was closed in 2019 (3 years prior to analysis)

) Methods, line 146 – weren’t all analyses retrospective?

) Data analysis, line 193 – continuous variables should be summarized with mean/SD when the variable has a normal distribution.

) Results – Table 1 – “key population” needs to be defined

) Discussion – lines 335-41. It is interesting, because in high income countries, high viral loads are considered more indicative of non-pill taking compared to resistant virus, as resistant virus is often less fit and is therefore associated with lower viral loads in the setting of ongoing partial suppression. It would be interesting to know if this finding was really dichotomous.

Please also note this comment from PLOS ONE: "PLOS ONE does not copyedit accepted manuscripts, so the language in submitted articles must be clear, correct, and unambiguous. Any typographical or grammatical errors should be corrected at revision, so please note any specific errors here."

(Reviewer comment to authors): There are too many errors to note each one with this initial submission. Prior to submission of the revision, please edit the manuscript with regard to spelling, grammar, language, and clarity).

6. PLOS authors have the option to publish the peer review history of their article (what does this mean?). If published, this will include your full peer review and any attached files.

Reviewer #1: No

Reviewer #2: No

---

## [Author Response · Author response to Decision Letter 0]

10 Jan 2023

Dear reviewers, 

We thank you for your time to review our manuscript. We have thoroughly revised the manuscript and feel it has strongly improved. 

In response to reviewer comments, the following major changes have been made to the manuscript: 

• We have clarified the populations used for different analysis throughout the document. 

• We have replaced table 1 with baseline characteristics for the overall population, stratified by group of access to viral load. 

• We replaced the cascade figure by a table comparing cascade results from those with a first viral load before near-POC introduction and after near-POC introduction

• We have added Kaplan-Meier analysis comparing the viral load access groups for mortality at 2 years. We also added analysis for both attrition and mortality, comparing only the groups who became eligible for viral load after viral load became available. 

• The manuscript was edited by an English language editor 

Please find below our point-by-point response to your suggestions. Since the manuscript has changed substantially, we refer to line numbers in the clean manuscript (Manuscript). We remain available for further queries and suggestions.

Reviewer #1: This was a very nicely written manuscript that touches on an important topic in a clear, structured way.

Thank you. 

Major Comments

1. The Abstract was a little unclear regarding the time of first viral load after initiation. Line 42-43 indicates 2.8 years after introduction of near POC VL, but then line 44 says 0.9 years in those becoming eligible after near POC VL. 3.7 years before near POC VL introduction. It would be helpful to clarify this as I’m guessing the intention was for the three different groups and/or one was an overall. Additionally, the results only include two of those three timings and would benefit from including all (lines 263-266).

We acknowledge this requires clarification. After thorough revision of the manuscript as indicated above, the abstract has been re-written. We have removed the timing from the abstract. We have compared time to first viral load between the two cascade groups (Table 2, Page 12). This comparison can be found on lines 254-255.

2. Please also review the patient numbers. The results were a bit unclear here. The abstract and results list 5271 PLHIV started ART between July 2009-December 2019. Line 230, however, lists 4291 PLHIV initiated ART. Then, Group 1 (794), Group 2 (2386), Group 3 (1111). Line 244 repeats 4291 initiated ART during the same timeframe and then indicates 3205 were eligible after near POC introduction (Group 3, earlier with n=1111). Finally, line 263, presents 945 eligible after near POC. It would be very helpful to ensure accuracy throughout and perhaps clearer explanations. The use of ‘Group 1’, ‘Group 2’, etc might also help.

Between July 2009 and December 2019, 5271 people started ART in the included clinics. However, 4291 (initiated ART between July 2009-June 2018) who became eligible for viral load before the start of 2019 were included. This population consists of three groups, based on which viral load platform was available when they became eligible for viral load: Group 1: No viral load (794), Group 2: Laboratory-based viral load (2388), Group 3: Near-point-of-care viral load (1109). We compared these three groups, as well as the two latter (new analysis) in terms of mortality (new analysis) and attrition. We have revised figure 3 into a new figure 2 on page 11-12 to reflect that. 

In a new table 2 on Page 12, we also provide a detailed viral load cascades of those who had a first viral load before and after POC viral load became available. This includes patients who were started in all different periods, but who remained in or entered the viral load cascade during the respective time periods. 

3. It would be helpful to discuss whether clinicians and facilities received more clinical training and/or viral load training refreshers at near POC VL implementation and that this could partly be a cause of the improvements seen.

Indeed, we have re-strengthened the knowledge on importance of regular viral load monitoring and management of high viral loads among clinical staff by doing several refresher trainings before near POC VL implementation. This is now reflected in the discussion section, lines 278-282. 

4. Some references could use an update. References 2 and 3 are on older WHO guidelines (2013 and 2016). These are referenced in the introduction and methods. These could be updated to 2021 as the referenced recommendations have all been carried forward and are the relevant, present guidelines.

The reference has been updated. 

5. The WHO treatment monitoring algorithm should be updated: testing is suggested at 6 months, 12 months, and yearly thereafter. The timing at 12 months post-ART initiation seems to be missing.

In practice, we perform viral load monitoring in line with the WHO treatment monitoring algorithm. This was in fact, a language mistake. The statement has now been corrected to reflect viral load monitoring at 6 months, 12 months and yearly thereafter in the introduction, lines 45-47. 

6. In the discussion, it would be helpful to touch on the possible role of near POC VL with uptake and implementation of DTG-based regimens, given the higher barrier to drug resistance and higher viral suppression rates. Further, please discuss how the current political situation in Myanmar could impact this strong system built.

After introduction of DTG-based regimens, compared to before, the proportion of patients with viral suppression is likely to increase. Near POC viral load would still allow to provide faster access to confirmation of a supressed viral load, and timely reaction in case of high viral loads. Among those with high viral loads, the proportion of high viral loads following non-adherence is likely to increase. Timely provision of EAC is also be facilitated by access to near POC viral load. This is reflected in the discussion section, lines 350-354.

However, in Myanmar, indeed, since these data were collected, services have been disrupted by a political crisis and the COVID-19 pandemic. The positive impact of the more robust regimens on viral suppression is likely thus be countered by unplanned treatment interruptions to poor health service access in this period. Effectively, a strike of health care staff in public hospitals, arrests of and threats to health care staff, and armed conflicts resulted in a gap in public health care. In addition, security issues restrict patients’ freedom to travel to facilities. However, near POC VL testing provides an opportunity as a mitigation measure to improve viral load testing access during these crises. Providing services faster and closer to patients by near POC VL testing could limit service interruption. This is elaborated in the discussion section, lines 383-391.

7. It would be helpful to discuss the previous NHL testing strategy. Why would they only accept 1-2 samples per clinic per month? Are they only testing one day a month? Is it due to staff, funding? This seems incredibly low and a waste of available resources.

Indeed, it was very low. This was because there were only two viral load platforms available for the whole country: one for upper Burma (NHL, Mandalay) and another one for lower Burma (NHL, Yangon). Since MAM sites are in Yangon in suburban slum areas, samples were sent to the lower Burma NHL machine, where they competed with all samples from all public facilities in the country’s South. Moreover, due to the limited availability of qualified staff, the machine was only utilised three days a week. Hence, we were only allowed to send 1-2 samples per month. This is elaborated in method setting, lines 117-123. 

8. In the discussion, please touch on data quality and the confidence you have in its accurateness, not just completeness

Patient files, including laboratory results, were entered manually by medical doctors and trained data clerks at the facilities were responsible for the transfer of data in the FUCHSIA database. The study database was thoroughly checked and corrected for discrepancies with source documents the principal investigator. We are thus confident the database reflects data collected from participants. Yet, data on key population status might not reflect the reality, as it is likely to be underreported. This is reflected in method section, lines 180-182 and discussion section, lines 367-369.

9. Please touch on how data were collected to ensure confidentiality – given that some patient identifying information was necessary to link across databases and files.

Immediately after testing positive for HIV at MAM clinics, all PLHIV received a number code for FUCHSIA, used to identify the person and link all medical files, including laboratory results. All team members use those codes when discussing case management or communicating viral load results. In FUCHSIA, no directly identifying information is recorded. In the study database, each patient was given a different study code. The key to link study codes with FUCHSIA codes are password protected, only accessible to the first author. This is explained in method section, lines 173-182.

10. In the discussion, it would be quite imperative to discuss the health of those initiating ART. Table 1 indicates that nearly 50% have advanced HIV disease and over 60% have WHO stage 3 or 4. Further, it was suggested that the majority of PLHIV in Myanmar identified as key populations, yet Table 1 has only 21%. Please explain and discuss this discrepancy.

We indeed had large proportions of patients with advanced HIV at presentation. Despite presentation with advanced disease, we did reach high proportions who were retained and had a supressed viral load; 95% after near POC introduction. While the proportion of patients presenting with advanced HIV decreased over time (Table 1, Page 10), it was still high in the most recent cohort, and it is likely to increase again seen the current situation in Myanmar. To reduce mortality in this group, a package of care including a POC test to detect CD4 counts below 200 cells/mm3 ,TB-Lipoarabinomannan and cryptococcal antigen is recommended. This is elaborated in discussion session, lines 287-290.

Indeed, in our cohort, only 19% of people identified as key population. Due to persisting stigma on key population groups, this is almost certainly a strong underestimation. We have recognised this in the discussion, lines 290-293.

To reduce mortality in this group, a package of care including a POC test to detect CD4 counts below 200 cells/mm3

11. For the results on lines 269-272 and in the discussion, it would be good to indicate that the value of the first viral load was the ‘only’ significantly associated metric of those assessed. Further, it would be good to include the OR in the text as well. Nearly 3x is important and significant.

We have mentioned it in result section, lines 259-262 and in discussion section, lines 341-343.

12. In lines 309 and elsewhere in the discussion, it may be important to note that it isn’t expected that a technology or device can do it all and/or impact adherence, retention, etc. Those must still be strengthened, etc.

Indeed, strengthening adherence throughout the treatment journey remains essential to avoid unsuppressed viral load and retain patients in care. Near-POC technology is just one among tools that could support adherence by providing a rapid turnover of viral load result. We have reworded this in discussion, lines 306-308 and lines 330-332. 

13. Line 326-327 is a little troubling, suggesting that adherence counseling not be necessarily for those with a first high VL. However, 36% of PLHIV in this study re-suppressed after adherence and likely were much better off staying on 1st line then switching unnecessarily. Further, this (EAC and switch hesitation) is likely to be particularly important with DTG scale-up.

We have reworded the discussion section to reflect these thoughts, lines 324-326 and lines 350-354. 

14. When discussing the recent WHO recommendation on POC VL (lines 356+), it may be useful to note that they are conditional based on high volumes and limited device throughput, particularly in high burden settings. To this point, it would be helpful to highlight the differences and challenges with POC vs near POC (ie same day versus not same day, need for clear follow-up structures and resources to maximize near POC use, possible improvements in TAT and clinical decision-making vs lab-based). Particularly as the model developed at these sites in Myanmar is clearly incredibly strong and well-resourced given impressive retention, etc.

Indeed, the usefulness of POC testing depends on the local HIV burden and the load of samples to be processed. We provided near POC VL testing, shared among three clinics serving a population living in suburban Yangon, an area with a general HIV-prevalence of 0.7%. Compared to POC, near POC is cheaper and easier to scale-up. Turnaround time in POC testing could be shorter compared with near-POC, and thus allow for faster case management. While we did not systematically collect turn-around times, experienced showed that most results were received by phone about two hours after the sample arrived at the testing site and on paper the next morning. We thus managed to provide prompt clinical action to high viral loads. In addition, we optimised utilisation of human and financial resources by sharing the instrument between three HIV clinics rather than one clinic. 

While POC VL testing in resource-limited high-burden settings can overcome major current challenges faced with centralized laboratory testing, high device cost and human resource needs limit a full scaling up POC viral load. Near-POC viral load might thus balance costs versus needs in medium-burden settings, with limited resources and access challenges, such as Myanmar. This is reflected in discussion section, lines 373-382. 

Minor Comments

1. Line 49, should this be ‘ninety-three’?

2. It would be helpful to reference lines 74-76.

3. For references on lines 82-83, it would be good to also reference Sacks et al: https://pubmed.ncbi.nlm.nih.gov/31274537/.

4. Line 101 is missing ‘in’ prior to ‘2017’.5. Line 127: Abbott should be written with two ‘t’s.

6. Line 163, it is unclear what ‘KAP status’ means.

7. Please look at line 193-194, it seems incomplete.

8. The sentence from line 208-210 seems irrelevant and could be removed.

9. Line 311, ensure inclusion of ‘near POC’.

10. Lines 329-330 should also reference MSF’s work on the VL cascade: https://msfaccess.org/sites/default/files/MSF_assets/HIV_AIDS/Docs/AIDS_report_Part_1_MakingViralLoadRoutine_MSF_VL_Programmatic%20Report_Web_2016_ENG.pdf (Table 1).

11. Lines 340-341 seemed a bit out of place with that paragraph.

12. Line 347, I might suggest ‘…generalizable to other public facilities.’

13. Line 360 likely needs an ‘and’ after ‘COVID-19’.

We have revised and edited the document to respond to minor comments 1-13. 

Reviewer #2: In this article by Tun et al., the authors describe implementation of near point-of-care quantitative HIV RNA testing in three clinics in Burma. The implementation of POC NAT for viral load monitoring is critical for HIV care and this manuscript presents important results. However, I found the analysis and presentation to be confusing and difficult to follow.

It is unclear why baseline characteristics are being presented only after other data and only for a subset of the population. The first results presented should be the baseline characteristics of the populations for all three time periods, divided first into people initiating ART during the three time periods.

We have revised table 1, which now presents baseline characteristics of the population starting ART between June 2009-July 2018, stratified by the three time periods in which there was different access to viral load. (Table 1, page10)

Then I would suggest the main analyses to be presented are:

1) How many total HIV RNA tests were done before and after implementation of POC NAT, and what was the number per for patient on ART (note that this will be different than the baseline groups, because some people will carry forward into the next time period).

We have added this relevant analysis in result section, lines 223-225. 

Before 2017, 432 viral loads were done among 2902 eligible patients (alive on ART for minimum six months and on ART six months), or 0.2 viral loads per person. 

After 2017, 7140 viral load tests were done among 3561 eligible patients, or 2.0 viral loads per person. 

What was the actual turn-around-time of POC NAT? 

I cannot find either of these results.

We did not capture the turn-around-time in a systematic way. However, experiences showed that most results were received by phone about two hours after the sample arrived at the testing site. The next morning, the paper result was sent to the facility. We have acknowledged this as a limitation and added it to the discussion section, lines 361-364. 

2) What % of eligible persons received their first VL testing in each of the three periods (<2013, 2013-2017, >2017)?

Among those eligible, 0% (0/732) had a first viral load before 2013, 14.6% (370/2532) between 2013 and 2017 and 91.9% (2945/3205) after 2017, respectively (Result section, lines 248-250). 

These are the data shown in Fig 2 and I believe in lines 219-224. If these are the same data, they should be labeled on the actual figure and can be removed from text.

These are not the same data. The previous figure 2 (now figure 1), we showed how many among those eligible had minimum one viral load in that year. We have clarified this in result section lines 225-228. 

Is there an explanation for why the number was so much lower in 2017 c/w 2019? That explanation should be included in the Discussion.

Before 2017, the numbers were extremely low. This was because there were only two viral load platforms available for the whole country: one for upper Burma (NHL, Mandalay) and another one for lower Burma (NHL, Yangon). Since MAM sites are in Yangon in suburban slum areas, samples were sent to the lower Burma NHL machine, where they competed with all samples from all public facilities in the country’s South. Moreover, due to the limited availability of qualified staff, the machine was only utilised three days a week. Hence, we were only allowed to send 1-2 samples per month. This is elaborated in method section lines 117-123 and further explained in discussion section lines 277-284. 

The comparison (line 223) should not be 2010-19 v 2013-19 v 2017-19, and there is probably a more informative statistical test that could be used here to show the magnitude of change.

After discussion with our statistician on the best test, we have added a logistic regression of year versus having a viral load done when eligible at the start of that year. Having a viral load done among those eligible was significantly associated with year: Odds Ratio: 5.21 [95% CI: 4.95-5.48]. The strength of the association did not change when excluding years without viral loads. (Result section, lines 228-230) 

3) Comparing the two time periods (2013-2017 and after 2017):

Of persons who had a first VL>1000, what % had a second test?

If they had a second test, what % had a second VL >1000

Of persons with a second VL>1000, what % had an ART change

Of persons with an ART change, what % had a follow-up VL < (under) 1000.

I do not see any data on the 2013-17 period.

We have added this analysis in table 2 (page 12), including a comparison of the two time periods, and used it to replace figure 4. 

4) Among people starting ART in 2017 who had access to POC NAT, was there lower mortality compared to historical controls (perhaps people starting ART in 2013, when VL testing became available).

We have performed additional Kaplan Meier analysis to compare: 

- 2-year attrition between group 2 and group 3

- 2-year mortality between the three groups 

- 2-year mortality between group 2 and 3 

We have created one figure (Figure 2, page 11-12) showing four panels with these analyses to replace the previous figure 3. 

This looks like it would be the data shown in Figure 3, although I am not clear on what definition of “attrition” is being used – is it discontinuation of ART? Transfer to another clinic? Lost to follow-up? Mortality? A combination endpoint of all of the above? This should be clarified in the Methods section. Mortality is, of course, the endpoint that matters most.

Attrition included those dead and lost-to-follow up. People transferred out were not excluded from analysis (Method section, Line 170)

It would be helpful to label the figure legend so that the reader can easily understand what Groups 1-3 represent.

We also clarified figure legends for all graphs.

Note that these analysis groups will be different from those in the above (persons eligible for first VL) because there will be survivor bias among persons who were started on ART in the prior period and who survived without NAT into the next period (presuming there is a universal recommendation to start ART regardless of CD4 count and that persons initiating ART in the two periods do not significantly differ in baseline characteristics)

The different analysis groups are specified in the method, lines 133-144. 

Minor comments:

1) I am not sure that this manuscript describes “uptake”, since it is not clear which patients were offered POC NAT and who declined.

We have changed the term to compliance with yearly viral load, throughout the document. 

2) Please note somewhere (probably in the methods – around lines 121-126) whether the ART start guidelines changed (e.g. based on CD4 versus universal testing) and whether HIV treatment options changed over the course of the study period, as improvements in ART would confound the analysis.

In Myanmar, universal treatment (i.e. regardless of CD4-count), was implemented after January 2017. Single-pill TDF-based regimens became available in November 2013 and DTG-based regimens in December 2018. We have shown this in the method (lines 108-110 and 113-114).

3) Methods, line 137 – please provide explanation for why the database was closed in 2019 (3 years prior to analysis). 

After database closure, the first author and main analysis, NNT, medical advisor of MAM, was strongly involved in mitigating impact of COVID-19 and military coup. We have acknowledged this is the discussion section (lines 383-385). 

4) Methods, line 146 – weren’t all analyses retrospective?

All analysis was retrospective, we have corrected at the lines 128-130, they now reflect that both the cross-section and cohort analysis were retrospective. 

5) Data analysis, line 193 – continuous variables should be summarized with mean/SD when the variable has a normal distribution.

We have corrected the data analysis section to reflect the use of means and standard deviations or medians with interquartile ranges, as appropriate. (Method section, lines 185-187)

6) Results – Table 1 – “key population” needs to be defined

We have defined key population in the legend of the new table 1, and added stratified analysis in the text (Result section, lines 216-220). This section reads; 

Overall, 541 people self-reported to be female sex worker (12.6%), 170 men-who-have-sex-with-men (4.0%), and 85 people-who-inject-drugs (2.0%). Among those included in group 1, 2 and 3 respectively; 53 (6.7%), 289 (12.1%), 199 (17.9%) reported to be FSW, 10 (1.3%), 77 (3.2%), 83 (7.5%) to be MSM and 8 (1.0%), 57 (2.4%) and 20 (1.8%) to be PWID.

7) Discussion – lines 335-41. It is interesting, because in high income countries, high viral loads are considered more indicative of non-pill taking compared to resistant virus, as resistant virus is often less fit and is therefore associated with lower viral loads in the setting of ongoing partial suppression. It would be interesting to know if this finding was really dichotomous.

We found indeed that a first high VL result of ≥5000 copies/ml was significantly associated with confirmed virological failure (adjusted odds ratio 2.61, 95%CI:1.02-6.65). Similar findings were also found in Zimbabwe and Ethiopia where PLHIV with a first VL of respectively ≥5000 copies/ml and ≥10,000 copies/mL also had higher odds of virological failure after EAC. Because of lack of drug resistance testing, it is difficult to ascertain whether the patients are failing because of adherence problems or because of resistance. One factor that could explain the difference between high- and low-income countries is the frequency with which viral load is measured and the types of drugs used. In the era of dolutegravir-based regimens, with high resistance barrier, it would be interesting to study this further, to see whether this factor continues to be important. We have reflected on this in the discussion, lines 341-354.

---

## [Decision Letter · Decision Letter 1]

3 Feb 2023

PONE-D-22-20673R1Near point-of-care HIV viral load testing: Cascade after high viral load in suburban Yangon, MyanmarPLOS ONE

Dear Dr. Tun,

Thank you for submitting your manuscript to PLOS ONE. After careful consideration, we feel that it has merit but does not fully meet PLOS ONE’s publication criteria as it currently stands. Therefore, we invite you to submit a revised version of the manuscript that addresses the points raised during the review process.

 Please pay close attention to the comments I have made concerning Reviewer 1's initial comments, as well as the new comments from Reviewer 2.

We look forward to receiving your revised manuscript.

Kind regards,

Julie AE Nelson, PhD

Academic Editor

PLOS ONE

Journal Requirements:

Additional Editor Comments :

From Reviewer 1 comments:

Adding “Group 1” type labels for the three groups was suggested and these labels are introduced in the methods on page 6. However, it would help Figure 2 to add the group names in the legends within the graphs (such as Group 1 No VL). The overall legend for Figure 2 needs more explanation of the difference between A and C, and the difference between B and D. Here is a suggested addition to this figure legend: Panels A and B include all three groups while Panels C and D only. compare Groups 2 and 3.

Table 3: add the explanation for “part of key population” to Table 3 that was used in Table 1 so the reader does not need to remember.

Reviewers' comments:

Reviewer's Responses to Questions

**Comments to the Author**

1. If the authors have adequately addressed your comments raised in a previous round of review and you feel that this manuscript is now acceptable for publication, you may indicate that here to bypass the “Comments to the Author” section, enter your conflict of interest statement in the “Confidential to Editor” section, and submit your "Accept" recommendation.

Reviewer #2: (No Response)

2. Is the manuscript technically sound, and do the data support the conclusions?

Reviewer #2: Partly

3. Has the statistical analysis been performed appropriately and rigorously? 

Reviewer #2: Yes

4. Have the authors made all data underlying the findings in their manuscript fully available?

Reviewer #2: Yes

5. Is the manuscript presented in an intelligible fashion and written in standard English?

Reviewer #2: Yes

6. Review Comments to the Author

Reviewer #2: In this manuscript, Tun et al present an updated report describing the implementation of POC VL testing in Burma. Thank you so very much for an extremely thoughtful revision to the original comments.

Seeing these data, I only have one comment:

In new Table 2, there are two main conclusions one could draw. The first is the very clear finding that more people received VL testing when POC testing was available. The second, however, is that people who had VL testing done and who had a VL>1000 copies - they were more likely to have documented suppression if their testing was done by the lab c/w POC testing - this is a bit counter-intuitive.

If I'm reading the data correctly there were 49-23= 26 (49%) people out of the 53 with >1000 copies in 2013-2017 who had a documented suppressed VL after EAC and only

125-84 = 41 (28%) out of 149 after 2017 -

p = 0.004.

Are there possible explanations for this?

7. PLOS authors have the option to publish the peer review history of their article (what does this mean?). If published, this will include your full peer review and any attached files.

Reviewer #2: No

---

## [Author Response · Author response to Decision Letter 1]

9 Feb 2023

Dear editor, reviewers, 

We thank you for your time to review our manuscript for the second time and provide comments for improvement. Please find with this “Response to reviewers” the following documents for your review: 

• The marked revised manuscript: 'Revised Manuscript with Track Changes'.

• The unmarked version: 'Manuscript'.

In this letter, we have provided our feedback to your comments and suggestions in blue.

Reviewer 1 

Adding “Group 1” type labels for the three groups was suggested and these labels are introduced in the methods on page 6. However, it would help Figure 2 to add the group names in the legends within the graphs (such as Group 1 No VL).

The new figure 2, with labels, is submitted with this rebuttal. 

The overall legend for Figure 2 needs more explanation of the difference between A and C, and the difference between B and D. Here is a suggested addition to this figure legend: Panels A and B include all three groups while Panels C and D only compare Groups 2 and 3.

We have replaced the legend of figure 2 with the following legend: 

Figure 2. Comparing outcomes on ART among PLHIV with different access to VL testing

Panel A, C present two-year attrition (dead, lost-to-follow-up) and panel B, D two-year mortality between PLHIV on ART who, when becoming eligible, had no access to VL testing (Group 1: No VL), access to laboratory-based VL testing (Group 2: Laboratory-based VL) or access to near POC VL testing (Group 3: Near POC VL). Panels A and B include all three groups while Panels C and D only compare Groups 2 and 3.

ART = antiretroviral therapy, PLHIV = people living with HIV/AIDS, POC = point of care, VL = viral load

Table 3: add the explanation for “part of key population” to Table 3 that was used in Table 1 so the reader does not need to remember.

This was added in the legend of table 3. 

Reviewer #2: 

In this manuscript, Tun et al present an updated report describing the implementation of POC VL testing in Burma. Thank you so very much for an extremely thoughtful revision to the original comments.

Thank you. 

Seeing these data, I only have one comment:

In new Table 2, there are two main conclusions one could draw. The first is the very clear finding that more people received VL testing when POC testing was available. The second, however, is that people who had VL testing done and who had a VL>1000 copies - they were more likely to have documented suppression if their testing was done by the lab c/w POC testing - this is a bit counter-intuitive.

If I'm reading the data correctly there were 49-23= 26 (49%) people out of the 53 with >1000 copies in 2013-2017 who had a documented suppressed VL after EAC and only 125-84 = 41 (28%) out of 149 after 2017 -p = 0.004.

Are there possible explanations for this?

Thank you. 

During the laboratory-based VL testing period, there was limited testing capacity. Hence, targeted VL testing prioritized PLHIV with demonstrated adherence problems, who could have had more benefit from the EAC sessions. In the near-POC era, back-log cases who had never had a VL were tested, who might have been failing for a long time. That could explain less re-suppression after EAC, due to true treatment failure. That could explain more patients receiving a follow-up VL result of ≤1000 copies/ mL after EAC in the laboratory-based testing era than in the near-POC era.

We have added a line in the results (lines 258-60 in Manuscript) stating: 

“Among PLHIV with a first VL>1000 copies/mL between 2013 and 2017, and those after 2017, respectively, 49% (26/53) and 28% (41/125) had a suppressed follow-up VL (p= 0.004).”

In the discussion, we have added (lines 334-40 in Manuscript): 

“Among PLHIV with a first high VL after near POC implementation, less had a suppressed follow-up VL documented compared to those who had a first high VL before. This could be explained by the fact that due to the limited laboratory-based testing capacity, targeted VL testing prioritized PLHIV with demonstrated adherence problems. This population could thus have benefitted more from the EAC sessions. In the near-POC era, all back-log cases who had never had a VL were tested, among whom more might have experienced true treatment failure”.

---

## [Editor Report · Decision Letter 2]

13 Feb 2023

Near point-of-care HIV viral load testing: Cascade after high viral load in suburban Yangon, Myanmar

PONE-D-22-20673R2

Dear Dr. Tun,

We’re pleased to inform you that your manuscript has been judged scientifically suitable for publication and will be formally accepted for publication once it meets all outstanding technical requirements.

Kind regards,

Julie AE Nelson, PhD

Academic Editor

PLOS ONE
---

## [Editor Report · Acceptance letter]

4 Apr 2023

PONE-D-22-20673R2 

Near point-of-care HIV viral load testing: Cascade after high viral load in suburban Yangon, Myanmar 

Dear Dr. Tun:

I'm pleased to inform you that your manuscript has been deemed suitable for publication in PLOS ONE. Congratulations! Your manuscript is now with our production department. 

Kind regards, 

on behalf of

Dr. Julie AE Nelson 

Academic Editor

PLOS ONE